# Study of Selective Dry Etching Effects of 15-Cycle Si_0.7_Ge_0.3_/Si Multilayer Structure in Gate-All-Around Transistor Process

**DOI:** 10.3390/nano13142127

**Published:** 2023-07-21

**Authors:** Enxu Liu, Junjie Li, Na Zhou, Rui Chen, Hua Shao, Jianfeng Gao, Qingzhu Zhang, Zhenzhen Kong, Hongxiao Lin, Chenchen Zhang, Panpan Lai, Chaoran Yang, Yang Liu, Guilei Wang, Chao Zhao, Tao Yang, Huaxiang Yin, Junfeng Li, Jun Luo, Wenwu Wang

**Affiliations:** 1Key Laboratory of Microelectronics Devices & Integrated Technology, Institute of Microelectronics of Chinese Academy of Sciences (IMECAS), Beijing 100029, China; liuenxu@ime.ac.cn (E.L.); tyang@ime.ac.cn (T.Y.);; 2Microelectronics Institute, University of Chinese Academy of Sciences, Beijing 100049, China; 3Beijing Superstring Academy of Memory Technology, Beijing 100176, China; chao.zhao@bjsamt.org.cn; 4Hefei National Laboratory, Hefei 230088, China

**Keywords:** GAA, selective etch, silicon germanium, etch effect, process simulation

## Abstract

Gate-all-around (GAA) structures are important for future logic devices and 3D-DRAM. Inner-spacer cavity etching and channel release both require selective etching of Si_0.7_Ge_0.3_. Increasing the number of channel-stacking layers is an effective way to improve device current-driving capability and storage density. Previous work investigated ICP selective etching of a three-cycle Si_0.7_Ge_0.3_/Si multilayer structure and the related etching effects. This study focuses on the dry etching of a 15-cycle Si_0.7_Ge_0.3_/Si multilayer structure and the associated etching effects, using simulation and experimentation. The simulation predicts the random effect of lateral etching depth and the asymmetric effect of silicon nanosheet damage on the edge, both of which are verified by experiments. Furthermore, the study experimentally investigates the influence and mechanism of pressure, power, and other parameters on the etching results. Research on these etching effects and mechanisms will provide important points of reference for the dry selective etching of Si_0.7_Ge_0.3_ in GAA structures.

## 1. Introduction

In integrated circuit manufacturing, engineers have been working hard to push devices to downscale critical dimensions in accordance with Moore’s Law [1]. The advanced technology node has adopted three-dimensional (3D) multilayer in the vertical direction after reaching the limit of integration in the horizontal direction. In terms of logic devices, the MOSFET structure has evolved from the conventional planar structure to the fin structure (FinFET), and then to the gate-all-around (GAA) structure in the past decade [2,3,4]. In terms of memory devices, 3D stacking can boost memory density and performance while reducing costs. The 3D-NAND relies more on stacked layers to break the bottleneck of memory capacity, the current maximum of which has exceeded 200 layers [5]. For DRAM memory, 3D-DRAM can follow the example of 3D-NAND flash memory by flipping the battery and stacking a large number of layers to meet the shrinking capacitor size. Samsung first demonstrated this structure for 3D-DRAM at the VLSI conference in June 2023 [6]. Since 3D-DRAM requires a large number of layer stacks to satisfy the space consumption in the horizontal direction, research on the etching of ultra-multilayer structures becomes necessary.

In order to achieve a multilayer structure in GAAFET and 3D-DRAM, it is necessary to employ the epitaxy technology of Si_1−x_Ge_x_/Si multilayer [7] and the selective etching of Si_1−x_Ge_x_ [8]. This paper will focus on the selective etching of Si_1−x_Ge_x_. The reason for the etching selectivity of SiGe is that the bond energy of Si–Ge is lower than the bond energy between Si–Si, making it easier to break [8]. Another explanation is that the doping of Si with Ge reduces the activation energy of the reaction, resulting in a reaction rate for Si_1−x_Ge_x_ greater than that for Si [9]. At present, conventional methods for Si_1−x_Ge_x_ selective etching mainly include wet etching [10], gaseous HCl etching [11], and dry etching [12]. In wet etching, a mixed solution containing H_2_O_2_, HNO_3_, CH_3_COOH, and HF is usually adopted to selectively etch the Si_1−x_Ge_x_ layer [13,14]. However, there are significant limitations in high-density circuit arrays and nanosheet devices with large aspect ratios. Vapor phase etching using HCl in chemical vapor deposition (CVD) reactors is also limited by its high-temperature environment and crystalline orientation-dependent etching [15]. Dry plasma etching has become a common method for etching Si_1−x_Ge_x_ in recent years, usually using plasma containing halogen elements for selective etching. Dry plasma etching mainly uses CF_4_ and NF_3_ as etching gases, together with Ar, O_2_, and He as auxiliary gases. The method can control critical dimensions well and has better etching uniformity, which is more desired in GAA devices and 3D-DRAM [16,17,18,19,20].

In order to obtain higher performance, multiple silicon channel layers can be stacked in the vertical direction [21]. Barraud successfully prepared a stack of seven silicon nanosheets, the driving current of which was three times that of two-cycle-stacked devices [22]. Seven or eight Si nanosheets is the limit reported in the literature so far. Stacks with more nanosheets, such as 10 or even 15 nanosheets, have not been studied in detail. Since SiGe selective-etching technology is an important key process in GAA devices, the articles published in recent years usually only show the process results, without mentioning the details of the process [20]. As for analysis of the etching effects, an article in 2019 analyzed the corner-rounding problem due to the diffusion of germanium elements caused by anisotropic etching, but rarely analyzed the profile after isotropic etching. Therefore, this paper focuses on the effects of multilayer etching and explores the future use of selective etching of SiGe in 3D-DRAM.

In our previous research work, we mainly focused on the Si_1−x_Ge_x_ selective etching of a three-cycle multilayer structure using CF_4_/O_2_/He gas [12]. In the balance process between stacked epitaxial growth and SiGe selective etching, Si_0.7_Ge_0.3_ is considered the optimal atomic ratio [23]. This study is the first to obtain a good isotropic Si_0.7_Ge_0.3_ selective-etching process using a conventional inductively coupled plasma (ICP) etching on a 15-cycle multilayer structure, and it also combines simulation with the process to optimize the phenomena that occur during etching and analysis. In a stack of more than a dozen nanosheets, the etching differences and mechanisms at different locations above and below are worth studying and discussing. Simulations by a commercial plasma process simulator named PEGASUS 2022 (PEGASUS Software Inc., Tokyo, Japan) show that random effects in etching are susceptible to the internal environment of the cavity in a multilayer. The edge of Si nanosheets will display asymmetrical damage phenomena on the up and down sides. Through the simulation, we investigated the influence of parameters such as pressure and power on the etching results of multilayers. And, as the number of stacked layers increases, the consistency of etch depth becomes an increasingly important challenge. By adjusting the process, we achieved a consistent etching of the 15-layer structure. Finally, we characterized the microscopic morphology of Si_0.7_Ge_0.3_ after etching, and discuss the layer quality and strain relaxation of SiGe during the removal process.

## 2. Materials and Methods

The experiments in this work were performed on 8-inch (100) silicon wafers. The Si_0.7_Ge_0.3_/Si multilayer was grown in an ASM E2000plus RPCVD reactor (ASM, Munich, Germany) with 15 cycles of Si_0.7_Ge_0.3_ (20 nm)/Si (20 nm) at around 650 °C. Before the epitaxial growth, a 200 mm Si (001) substrate was cleaned with DHF (1:100). Then, the wafer was loaded into the reduced-pressure chemical vapor deposition (RPCVD) chamber and baked at 1050 °C for 2 min to remove the native oxide. The main precursors were germane (GeH_4_) and dichlorosilane (SiH_2_Cl_2_), respectively. And the Si_0.7_Ge_0.3_/Si multilayer was grown with H_2_ as the carrier gas. Then, a hardmask was deposited on top of the epitaxial layers. Photolithography was carried out to define the pattern in the photoresist on top of the hardmask. Both anisotropic and isotropic etching in this experiment were carried out using an 8-inch ICP etching tool Lam9400DFM (Lam Research Inc., Fremont, CA, USA), and hardmask openings were formed using CF_4_/HBr/O_2_ gas. HBr/O_2_ plasma was used to vertically etch the stacked Si_0.7_Ge_0.3_/Si multilayer film to obtain a mesa structure with a width of 1 μm. Afterwards, the prepared samples were cut into slices of about 3 × 3 cm^2^ to facilitate etching experiments. Finally, we used CF_4_, O_2_, and He for the selective etching of the Si_0.7_Ge_0.3_. The temperature was set to 80 °C, the pressure range was 5–80 mT (full scale 80 mT), the source power range was 200–800 W (full scale 800 W), and bias RF was set to 0 W to minimize ion bombardment. The experimental process is shown in Figure 1.

PEGASUS 2022 is capable of simulating the time-dependent feature profiles that result from multiple physical and chemical reactions. It employs the Monte Carlo method for the simulation to model the reflection, deposition, etching, and sputtering reactions and the corresponding reaction probabilities. We calculated two-dimensional (2D) profiles of the plasma etching using the feature profile simulation module (FPSM) of the PEGASUS 2022 software. Firstly, the geometry of the substrate to be etched is characterized by the volume occupancy of the solid layer. Second, it is important for the accuracy of our simulations to define as accurately as possible the physical and chemical reactions between the relevant particles (ions and neutral radicals) and the solid layer. When the occupancy becomes 0.0, the cell becomes empty (gas phase), which means it is etched away. This process is repeated until a preset time or all particles are consumed. Finally, the etching profile, gas density, and other important parameters are obtained.

## 3. Results and Discussion

### 3.1. Microscopic Characterization of Epitaxial Thin Films

The epitaxy of multicycle Si_0.7_Ge_0.3_/Si determines the number of future conductive channels and is the basis for the preparation of channel quality together with channel release etching. In this paper, using epitaxy, 15-cycle multilayer structure of Si_0.7_Ge_0.3_/Si was alternately deposited, each layer with a thickness of approximately 20 nm. As shown in Figure 2a, the interface between the Si_0.7_Ge_0.3_ and the Si is clear, the thickness of each layer is uniform, and the interface profile is straight. Figure 2b is an energy-dispersive spectroscopy (EDS) line-scan diagram of the epitaxial region. The results show that among the 15 Si_0.7_Ge_0.3_ layers, the Ge content of each layer is constant at around 30%, and the boundaries of each layer are consistent with the growth of the material. They also show that there is no obvious interdiffusion at the interface of different interlayer materials at the junction of the epitaxial stacks. The above results all show that a high-quality 15-cycle Si_0.7_Ge_0.3_/Si has been prepared by epitaxy, laying the foundation for subsequent selective etching.

### 3.2. Simulation

PEGASUS 2022 simulation was used to obtain the results of the etching profile under different conditions. By constructing the ICP cavity environment, CF_4_, O_2_, and He were used to simulate the selective etching of the Si_0.7_Ge_0.3_/Si. In assessing the results of multilayer structure, the uniformity of etching is an important evaluation index. Figure 3a shows that the profile of the 15-cycle was affected by the gas pressure. Figure 3b shows the range of etching depths for the 15 Si_0.7_Ge_0.3_ layers under different gas pressures. With the optimization of internal gas uniformity, the etching depths of different layers became more and more consistent, an effect which was also quantified using standard deviation, with the standard deviation value decreasing from 24.98 to 9.21. This random effect may be related to the result of gas diffusion, which leads to the inequal diffusion between the by-products after etching and the etching gas.

In order to truly simulate the ICP etching process, the etching model must include physical (ions) and chemical (radicals) etching processes. In chemical etching, free radicals diffuse to wrap the whole surface area of the sample profile, which is completely isotropic. However, as shown in Figure 4, there is some damage on the Si layer post-etch simulation, and the damage is more severe on its upper surface and is asymmetrical. It is noteworthy that this damage is also mainly concentrated at the nanosheet edges, while the internal profile looks better. Incident ions may be the root cause of this asymmetric damage. The ions are directional after colliding with each other, but most of the damage is concentrated on the upper surface of the Si nanosheets due to the incident angle. In the follow-up experiments, we will conduct an in-depth exploration and analyze the impact of the random effect and the etching damage.

### 3.3. Pressure Impact on Isotropic Dry Etching

To verify the simulation results and the impact of pressure on etch performance, we conducted experiments at different gas pressures using ICP. Based on the above-mentioned simulation, we found that the pressure had a great influence on the random effect in the multilayer. Experiments with different pressures were conducted, and the etching results are shown in Figure 5. In the high-pressure range, as shown in Figure 5a, the randomness is relatively large. Even under the same conditions, the etching depth of the Si_0.7_Ge_0.3_ layer at the same position in the two experiments is not completely repeatable. The etch uniformity is gradually improved as the pressure is reduced, as shown in Figure 5b. As shown in Figure 5c, the etching profile is quite uniform at low pressure, indicating that the etching depth of each Si_0.7_Ge_0.3_ layer is consistent, which can be seen in Figure 5d.

This phenomenon may be attributed to the result of gas diffusion in the cavity [24]. During the etching process, the lower pressure means that the by-products of the etching are desorbed from the cavity-bottom surface and pumped out at a fast rate and do not significantly hinder the diffusion of reactive gases into the bottom of the cavity. However, under high pressure, the rate at which by-products are drawn out of the cavity slows down, which hinders the diffusion of some etching gases, resulting in random etching effects. This experimental observation is consistent with the previous simulation results, indicating that the uniformity of the gas inside the cavity is affected by the pressure. By adjusting the pressure, the most consistent results after etching of the 15-cycle multilayer structure resulted in a standard deviation ϭ of 2.06, even better than that of the simulations. In relation to the structure of the seven-levels-stacked nanosheets published in 2020 [21], the result of SiGe selective etching is comparable, and the consistency is greatly improved.

In addition, we set the chamber condition at low pressure ranging from 5 mT to 50 mT and studied how the pressure influenced the Si_0.7_Ge_0.3_ etching rate and selectivity. As shown in Figure 6, the etching rate corresponding to the right axis is calculated by dividing the measured etching depth by a fixed etching time of 20 s. The plot shows that in the low-pressure regime, the etching rate is faster with increasing pressure. This trend can be explained that, at low pressure, the radical density is lower, resulting in a relatively low etching rate. When the chamber pressure is higher, the plasma density in the reaction chamber increases, more plasma radicals chemically react with the surface of the material, and the etching rate increases. In the pressure range of 5 mT–50 mT, the etch rate increases from 0.90 nm/s to 19.45 nm/s, and it can be seen that the etching rate is very sensitive to pressure changes. According to the fitting curve, the etching rate shows a quadratic relationship with increasing pressure.

Due to the different etch depths, it is difficult to directly compare the etch profiles under different conditions. We adjusted the etching time under different pressures and fixed the etching depth at around 80 nm. This is because the depth of 80 nm can meet the channel release in all current GAA devices as well as the selective etching of SiGe in 3D-DRAM. Etch selectivity was also calculated and corresponds to the left axis in Figure 6. The selectivity of the Si_0.7_Ge_0.3_/Si increases continuously with increasing pressure. In the case of lower pressure, the average free path of particles is longer, and the incident energy is higher. Both physical and chemical etching effects are more pronounced. As a result, the etching edge is severely damaged, and the selectivity is lower. As the pressure increases, ion collisions intensify, resulting in a decrease in particle incident energy and thereby reducing the effect of physical etching and improving the selectivity. However, as the by-product is not drawn out of the chamber in time, the edges of the structure become rough. By balancing the selectivity and consistency, the pressure condition was fixed at 20 mT in subsequent experiments.

### 3.4. Source Power Impact on Isotropic Dry Etching

There are two power sources in the ICP equipment: one is ICP source power and the other is RF bias power. Bias power controls the kinetic energy of particles accelerated to the surface through the electric field, causing the upper-surface damage in the simulation. Its directionality can lead to some degree of anisotropic etching. In order to reduce ion energy to weaken surface physical damage, the RF bias power was turned off. The plasma process was then working in a downstream-like mode.

In order to better understand the role of source power in CF_4_/O_2_/He plasma, we studied the relationship between the Si_0.7_Ge_0.3_ etching rate and the power conditions in the etching chamber. As shown in Figure 7, the etching rate corresponding to the right axis is calculated by dividing the measured etching depth by a fixed etching time of 20 s. When the power in the cavity is set in the range of 200–600 W, the etching depth or etching rate increases with the increase in power, and the etching rate increases from 0.94 nm/s to 4.41 nm/s. When the power in the cavity is set in the range of 600–800 W, the etching rate tends to reach saturation and stabilizes at about 4.4 nm/s. Overall, according to the fitting curve, the etching rate has a tendency to be linear and then gradually saturated with increasing power. This is because, as the ICP source power increases, the gas ionization rate increases, and the plasma density increases. Source power generates high-density plasma through inductive coupling, which determines the conversion rate of F radicals and the density of plasma. The increase in reactant ions enhances the chemical reaction and increases the etching rate. But as the source power continues to increase, the plasma density in the reaction chamber tends to saturate, and the chemical reaction between ions and the surface of the material to be etched reaches its peak.

In order to explore the influence of different source powers on the Si_0.7_Ge_0.3_/Si selectivity, we adjusted the etching time under different powers and fixed the etching depth at about 80 nm. Etch selectivity was also calculated and corresponds to the left axis in Figure 7. When the power is in the range of 200–400 W, the lower the power, the greater the Si loss. This is due to the fact that O_2_ requires higher power to dissociate compared to CF_4_, and the degree of dissociation is lower at low power. And the selectivity of Si_0.7_Ge_0.3_/Si is very sensitive to the O_2_ content, which leads to serious damage to the Si at low power. When the power is in the range of 500–800 W, the selectivity increases with the increase in power, the etching uniformity is better, the etching outline is rectangular, and the angle is relatively sharp. Generally speaking, when the power is in the range of 600–800 W, the etching selectivity reaches a relatively optimized condition.

### 3.5. Asymmetry Effect of Etching Damage

In order to more accurately characterize the process results in this study, the 15-cycle multilayer of Si_0.7_Ge_0.3_/Si after selective etching was characterized using TEM. The etching result is shown in Figure 8a. The etching amount of each Si_0.7_Ge_0.3_ layer is almost equal, with a relatively good uniformity. However, HRTEM results indicate the loss at the upper edge of the nanosheets under conditions of 20 mT and 600 W, resulting in significant silicon damage. It can be inferred from the image that the damage is mainly concentrated on the upper surface of the Si nanosheet edges, which is highly consistent with the phenomenon in the previous simulation.

The root cause of the asymmetric damage is described in Figure 8b. The wafer is grounded and allows positively charged ions to bombard the top surface almost vertically under self-bias. As a result, a portion of particles will bombard the edge of the Si nanosheets due to scattering by collision. During the etching of the Si_0.7_Ge_0.3_, the upper surface of the nanosheet is more easily damaged, while the lower surface of the nanosheet is significantly less damaged due to the shadowing effect. When we define the angle between the incident direction of the particle and the vertical direction as θ, the incident angle of the particle obeys a normal distribution. In other words, the number of particles with large θ is small. Therefore, the damage on the upper surface is mainly concentrated at the edge, which is an important reason for the selective etching of the Si_0.7_Ge_0.3_ using ICP being limited.

### 3.6. Micromorphological Characterization and Material Quality Analyses

Due to the relatively large number of cycled layers, residual by-products or diffusion of elements may occur after etching, so we performed elemental analysis of the etched structure. Figure 9 shows the elemental analysis of a cross section of the sample after etching. The EDS result images show that the boundary of each layer is consistent with the growth of the material, and there is no diffusion and accumulation of Si and Ge elements within the stack after etching. There are no intermediate products containing Ge elements on the surface and no polymers containing C elements that may remain after the use of CF_4_ gas. In addition, element C is the loading filler material in the TEM sample, and the oxide layer at the edge of the etched profile is the natural oxide layer formed after the contact of the sample with air.

To further analyze the state of the silicon after the etching process, we performed HRXRD characterization of the samples after epitaxy, anisotropic etching, and isotropic etching, respectively. To determine whether our samples were strained or relaxed, an asymmetric scan of (113) facet was required. Figure 10a shows that the SiGe peaks are consistent with the Si peak in the vertical direction, which implies that the epitaxial Si/SiGe multilayer is totally strained. Figure 10b,c show the Si and SiGe peaks after vertical etching and lateral etching, respectively. After vertical etching, the SiGe peaks shift away from the Si peaks, which shows the strain relaxation in the SiGe film [25]. This phenomenon is different from the results for the three-cycle structure in our previous experiments [12], indicating that anisotropic etching is more likely to cause relaxation problems as the number of stacked layers increases. Moreover, this problem still exists after lateral etching. From the experimental results so far, the etching rate of the 15-layer structure is slightly higher than that of the three-layer structure, while the selectivity ratio decreases [12]. We will continue to investigate the influence of the variation of stress on the etching results.

## 4. Conclusions

This paper investigates the use of a conventional ICP etching system for quasi-isotropic etching of a 15-cycle Si_0.7_Ge_0.3_/Si multilayer structure. By conducting simulation and experiments, the study reveals that the selective etching process in the Si_0.7_Ge_0.3_/Si multilayer structure causes random effects on the Si_0.7_Ge_0.3_ layer etch depth and asymmetric damage on the Si surfaces due to radical and ion distribution. Pressure was found to be the main factor for mitigating the random effect, and the standard deviation of the etching depth was reduced by more than 85% to 2.06 by lowering the pressure. For Si nanosheet edge damage, the upper-surface loss was about 2.3 times that of the lower surface; this phenomenon can be explained by bombardment of incident ions accelerated by self-bias. Finally, for a 15-cycle multilayer structure of Si_0.7_Ge_0.3_/Si with each layer having a thickness of 20 nm, good etch uniformity and a smooth surface were obtained. The selectivity of etching Si_0.7_Ge_0.3_ to Si was calculated to be 34 under conditions of 50 mT and 600 W, and an etching rate of 0.90 nm/s~19.45 nm/s was achieved through tuning the process conditions. And it was found that the 15-layer structure was more prone to relaxation in etching than the three-layer structure. In the future, the avoidance of random effects by using low pressure and filtering charged particles during the etching process will offer more application prospects.

## Figures and Tables

**Figure 1 nanomaterials-13-02127-f001:**
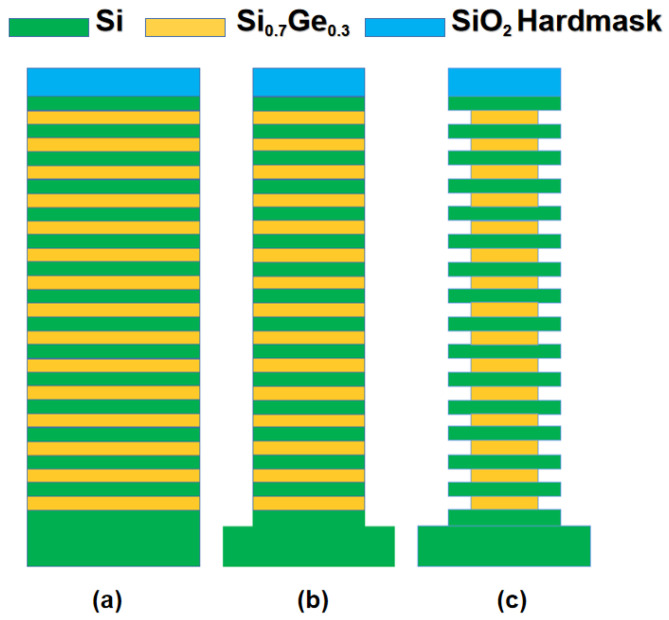
Process flow and schematic diagram of the experiment: (**a**) epitaxy Si_0.7_Ge_0.3_/Si multilayer structure; (**b**) photolithographic patterning and dry anisotropic etching; (**c**) dry selective isotropic etching.

**Figure 2 nanomaterials-13-02127-f002:**
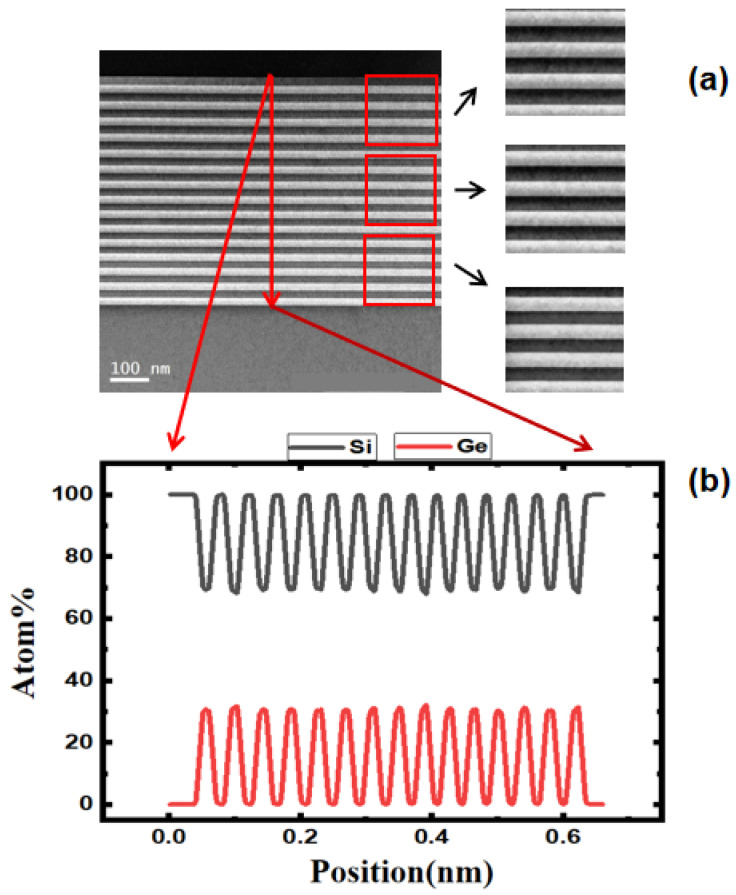
Transmission electron microscope (TEM) image of 15-cycle multilayer structure and EDS analysis: (**a**) high-resolution transmission electron microscope (HRTEM) of 20 nm Si and Si_0.7_Ge_0.3_ layer; (**b**) EDS line-scanning of filmstack.

**Figure 3 nanomaterials-13-02127-f003:**
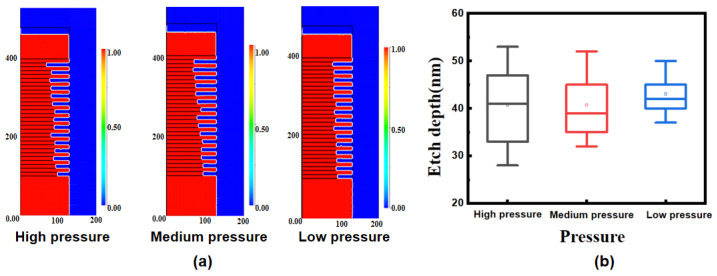
(**a**) Variation of random etching effects in S_i0.7_Ge_0.3_/Si multilayer structure obtained by PEGASUS 2022 simulation; (**b**) box plot of etching depth of 15 Si_0.7_Ge_0.3_ layers under different pressure conditions.

**Figure 4 nanomaterials-13-02127-f004:**
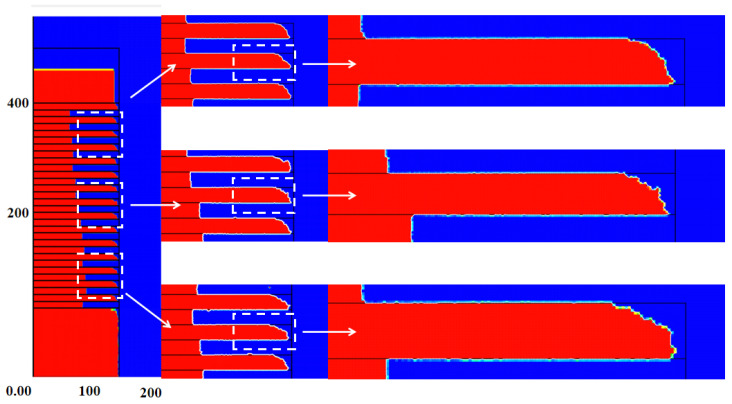
Asymmetry of etching on upper and lower surfaces from PEGASUS 2022 simulation.

**Figure 5 nanomaterials-13-02127-f005:**
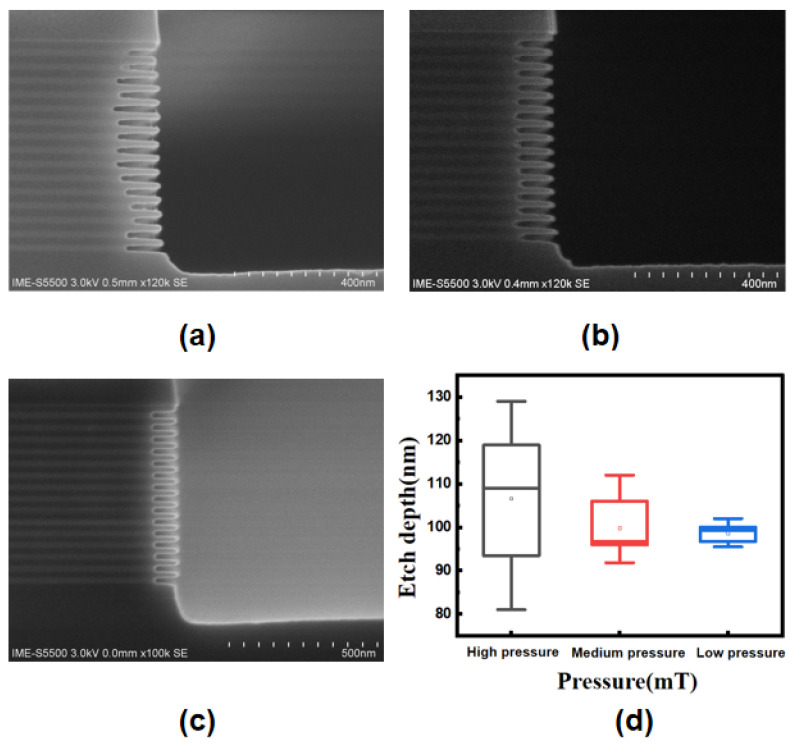
(**a**) Etched profile with high pressure; (**b**) etched profile with medium pressure; (**c**) etched profile with low pressure; and (**d**) box plot of etching depth for 15 Si_0.7_Ge_0.3_ layers under different pressures.

**Figure 6 nanomaterials-13-02127-f006:**
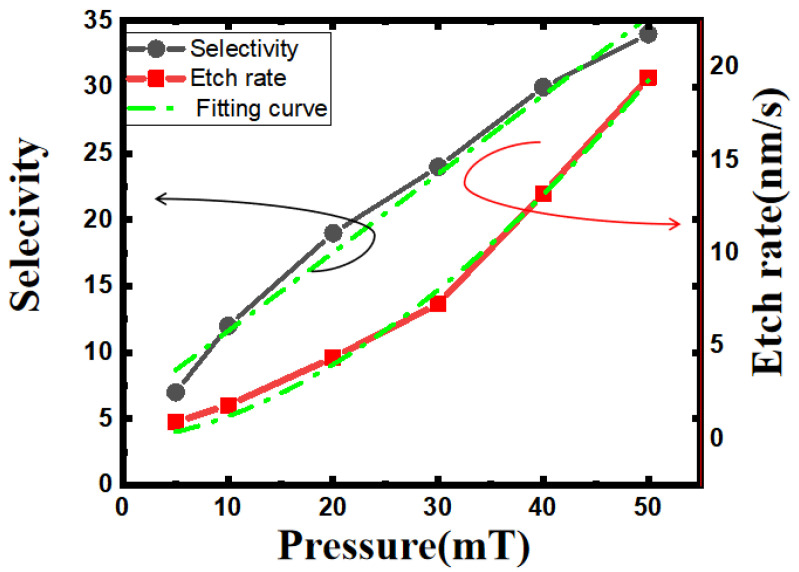
Trend of Si_0.7_Ge_0.3_ etch rate and selectivity in terms of increasing pressure.

**Figure 7 nanomaterials-13-02127-f007:**
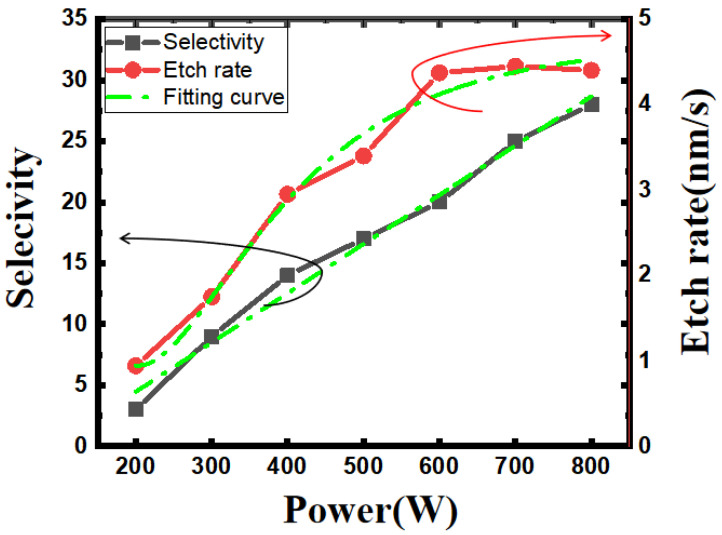
Trend of Si_0.7_Ge_0.3_ etch rate and selectivity in terms of increasing source power.

**Figure 8 nanomaterials-13-02127-f008:**
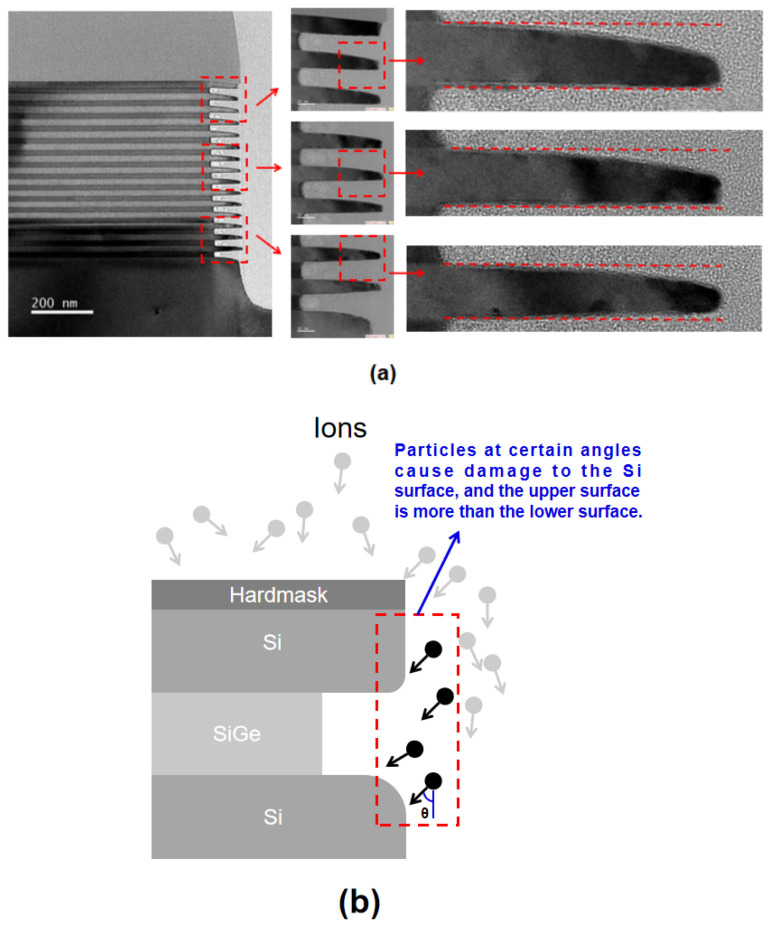
(**a**) TEM images of isotropic etching profile; (**b**) schematic diagram of top surface etch damage.

**Figure 9 nanomaterials-13-02127-f009:**
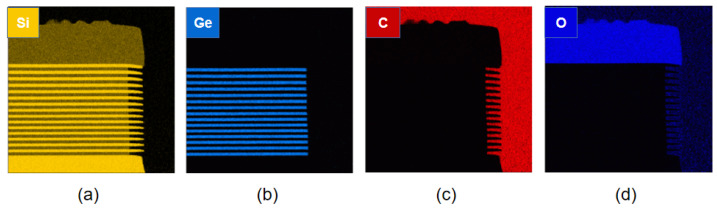
EDS mapping of multilayer structure post-isotropic etching: (**a**) scanning of silicon element of full map; (**b**) scanning of germanium element of full map; (**c**) scanning of carbon element of full map; and (**d**) scanning of oxygen element of full map.

**Figure 10 nanomaterials-13-02127-f010:**
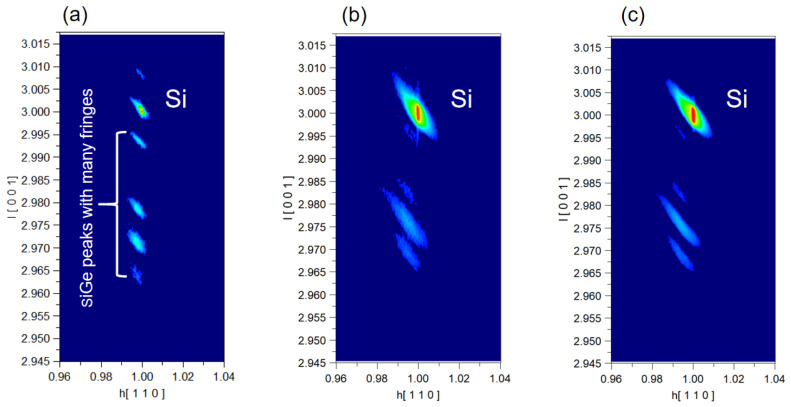
RSMs in the vicinity of the asymmetric (113) Bragg reflection acquired on SiGe/Si multilayer structure: (**a**) unprocessed structure; (**b**) after vertical anisotropic etch and 100:1 DHF wet clean; and (**c**) after SiGe isotropic selectivity etching.

## Data Availability

The data can be obtained from the authors upon request.

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
