# Peer review of "Study of Selective Dry Etching Effects of 15-Cycle Si0.7Ge0.3/Si Multilayer Structure in Gate-All-Around Transistor Process"

_nanomaterials, 2023, doi:10.3390/nano13142127_

Round 1

Reviewer 1 Report

In the present manuscript, Liu et al. have studied and optimized the selectivity of dry etching for SiGe/Si stacks in a gate all around process. The authors with the help of simulation and practical optimization were able to figure out that the asymmetry of edge etching could be significantly reduced by lowering the chamber pressure for fixed etch conditions to minimize scattering of plasma radicals at eh top edge surface. They further optimized the source power to etch 80nm edges and showed that ion exposure is significantly higher at the top surface with a gradient from the top to bottom edges due to a shadow effect and improved selectivity of SiGe to Si was observed at a higher power. The work is certainly interesting to the Nanomaterials community, especially to the vertical transistor community. I can recommend this manuscript for publication with some minor revisions.

1.       Can the authors briefly discuss the etching effects when the stoichiometry of the Si and Ge are changed to understand their individual sensitivity?

2.       Please explain why a depth of 80 nm specifically was chosen for the etch optimization.

3.       The introduction needs to discuss other similar approaches or materials where etching has faced some hurdles, establishing the novelty of their application.

4.       The results and discussions part should contain more comparisons with current literature and improved discussions with relations plasma reaction with the different material species. Some insights into the chemical bond energetics would be helpful in terms of obtaining selectivity.

Author Response

请参阅附件

Reviewer 2 Report

The paper titled "Selective Etching of Si1-xGex/Si Multilayer for GAA Devices and 3D-DRAM" discusses the challenges and advancements in the selective etching process of Si1-xGex/Si multilayer structures grown by RP-CVD.

The main weakness is related to the description of the software used for etching simulation. It is not clear what are the phenomena that are considered and thus it should be discussed more in detail.

The sentence “The epitaxy of multi-cycle Si0.7Ge0.3/Si is the key technology to determine the quantity and quality of future conductive channels. The preparation of high-quality Si0.7Ge0.3/Si multilayer is the prerequisite for achieving selectivity etching” present a slightly confusing perspective on the goals of the paper. The preparation of a high-quality multilayer serves as a prerequisite for developing high-quality devices, rather than primarily demonstrating an etching protocol. I would suggest to rephrase the sentence.

After some revision the paper could be valuable for researchers and engineers working in the field

I recommend to check the paper for typos

Reviewer 3 Report

Authors report about their investigation of selective dry etching of SiGe layers in multilayer Si0.7Ge0.3/Si stacks aiming to create the suitable structures for GAA FET fabrication. Selective dry etching (with subsequent inner spacer formation) is the important step in the process flow and so the related studies are of interest.

However, there are several questions to address.

1) My major concern is the novelty of the presented manuscript. Indeed, this topic was studied by authors previously and they have published a number of related works. For example, authors mentioned their previous works published in Nanomaterials (Ref. 2) and Journal of Materials Science: Materials in Electronics (Ref. 12). In particular, in the latter work authors have already found the suitable conditions for selective dry etching. The main difference of the current work, as authors mentioned by themselves, is the increased number of layers in the stack (from 3 SiGe/Si pairs in Ref. 12 to 15 pairs in this work). It seems doubtful that the increase of layer quantity will bring enough novelty.

Moreover, the silicon nanosheet damage after ICP etching which is thoroughly examined in the current manuscript could already be found in their previous work, see Fig. 9 in Ref. 12.

The random etching effect, that is another feature claimed as “novel” in the current manuscript seems to be more technical rather than physical issue.

To finalize, I suppose that the manuscript novelty is not sufficient. 

2) The epitaxial growth conditions of the SiGe/Si stack should be described. This is important because both each 20 nm thick Si0.7Ge0.3 layer and the structure 15x(Si0,7Ge0.3/Si) as a whole experiences relatively large elastic strain and at some conditions this strain could relax. See for example, Fig. 4 in [K. Brunner, Rep. Prog. Phys. 65, 27 (2002)]  - one can see that 20 nm thick Si0.7Ge0.3 layer is at least in the metastable state. 

3) In view of the previous remark it is highly desirable to provide the X-ray diffraction characterization of the fabricated structures. Besides, in the author’s previous work (Ref. 12) such an analysis was carried out. 

There are also some minor/technical remarks:

 4) What do the "fitting curves" in Figures 6 and 7 mean?

5) The two parts of Figure 8b are almost identical. Authors can leave only the right side of this figure.

6) English language deserves improvements. A number of grammar mistakes are present in the manuscript.

7) References are given in different styles, they should be unified

 Moderate editing of English language required

Round 2

Reviewer 3 Report

Authors have addressed the main points I’ve mentioned and so improved their paper. Nevertheless I suggest doing some additional work in order to make the manuscript more convincing. 

1) I suppose that it will be suitable if authors would highlight the novelty of their manuscript better. Some of the explanations presented in their cover (response) letter could be inserted in the manuscript text.

2) It would be better to include also the temperature of SiGe/Si layers epitaxy in order to clarify the growth conditions in more detail.

3) Authors have provided the XRD analysis and found the differences in strain relaxation between 3-fold and 15-fold SiGe/Si stacks. In particular, strain relaxation possibly occurs after etching of 15-fold stacks. Would this relaxation affect the etching rate and/or selectivity ? A brief discussion concerning this point would be beneficial.

4) The ref. 25 inserted in the following sentence “To further analyze the state of the silicon after the etching process[25],…” seems to be in somewhat wrong place. I suggest that ref.25 would be better suited somewhere below, closer to the discussion about the strain in SiGe films.

English should be checked once again.
